# Enhancing Clinical Utility: Utilization of International Standards and Guidelines for Metagenomic Sequencing in Infectious Disease Diagnosis

**DOI:** 10.3390/ijms25063333

**Published:** 2024-03-15

**Authors:** Chau-Ming Kan, Hin Fung Tsang, Xiao Meng Pei, Simon Siu Man Ng, Aldrin Kay-Yuen Yim, Allen Chi-Shing Yu, Sze Chuen Cesar Wong

**Affiliations:** 1Department of Health Technology and Informatics, The Hong Kong Polytechnic University, Hong Kong, China; kantrevor@gmail.com (C.-M.K.); andy_thf@yahoo.com.hk (H.F.T.); 2Department of Applied Biology & Chemical Technology, The Hong Kong Polytechnic University, Hong Kong, China; xiaomeng.pei@connect.polyu.hk; 3Department of Surgery, Faculty of Medicine, The Chinese University of Hong Kong, Hong Kong, China; simon.ng@cuhk.edu.hk; 4Codex Genetics Limited, Shatin, Hong Kong, China; aldrinyim@codexgenetics.com (A.K.-Y.Y.); allenyu@codexgenetics.com (A.C.-S.Y.)

**Keywords:** infectious disease diagnosis, next generation sequencing, metagenomics sequencing, implementation and validation, ISO15189, ISO24420, ISO20397

## Abstract

Metagenomic sequencing has emerged as a transformative tool in infectious disease diagnosis, offering a comprehensive and unbiased approach to pathogen detection. Leveraging international standards and guidelines is essential for ensuring the quality and reliability of metagenomic sequencing in clinical practice. This review explores the implications of international standards and guidelines for the application of metagenomic sequencing in infectious disease diagnosis. By adhering to established standards, such as those outlined by regulatory bodies and expert consensus, healthcare providers can enhance the accuracy and clinical utility of metagenomic sequencing. The integration of international standards and guidelines into metagenomic sequencing workflows can streamline diagnostic processes, improve pathogen identification, and optimize patient care. Strategies in implementing these standards for infectious disease diagnosis using metagenomic sequencing are discussed, highlighting the importance of standardized approaches in advancing precision infectious disease diagnosis initiatives.

## 1. Introduction

Infectious diseases caused by viruses, bacteria, parasites, or fungi, such as hepatitis B and C, diarrheal diseases, HIV/AIDS, tuberculosis (TB), respiratory infections, pneumonia, condylomas, etc., can be transmitted directly or indirectly from one person to another [1]. For many years, medical research and public health efforts have focused on identifying and managing infectious diseases. However, the understanding and management of infectious diseases have evolved. In the 20th century, molecular diagnostic technologies, such as polymerase chain reaction (PCR) and other DNA-based methods, such as in situ hybridization and Southern blot analysis, were used to identify specific RNA or DNA sequences of infectious agents and revolutionized the field of infectious disease diagnosis [2,3]. The utilization of next-generation sequencing (NGS) technology, also known as metagenomics sequencing (mNGS), in combination with bioinformatics, has also enabled researchers to identify pathogenic microorganisms more rapidly and accurately, compared with the conventional approaches, such as culture-based methods and serology methods like enzyme-linked immunosorbent assay (ELISA) [4,5,6]. In addition, genomic analyses facilitated by NGS can offer valuable insights for viral quasispecies analysis [7,8,9,10,11,12], and for genomics-informed outbreak investigations, including epidemic dynamics, source, and timing, and the geographical origin of pathogens [13,14,15,16,17,18,19,20]. By analyzing the genetic information obtained through NGS, researchers can gain a deeper understanding of the transmission patterns and genetic diversity of infectious agents, aiding in the development of effective control and prevention strategies [21,22,23,24].

Clinical mNGS has proven to be an effective diagnostic tool for infectious diseases due to its ability to detect a large range of pathogens. Recent advances in technology have made it an invaluable diagnostic tool for clinicians. mNGS is particularly useful for detecting infectious agents in cases of encephalitis and meningitis [25,26]. As an example, some studies have demonstrated the potential of mNGS in the diagnosis of meningitis and encephalitis. Fan et al. have demonstrated that by utilizing mNGS, infectious diseases caused by pathogens that are less frequently encountered, such as *Angiostrongylus cantonensis*, can be accurately diagnosed, which in turn results in improved patient outcomes and care [26]. Wilson et al. also demonstrated the comprehensive spectrum of potential causes that can be identified by a single mNGS assay, including viral, bacterial, fungal, and parasitic pathogens, showcasing its utility in diagnosing complex central nervous system infections [25]. Clinical mNGS has also shown promise in diagnosing various infectious diseases, including tuberculosis, HIV, and coinfections [27]. Wang et al. (2019) explored the potential of mNGS in identifying pathogens causing tuberculous meningitis, highlighting its potential to enhance tuberculosis-related infection diagnosis [27]. In addition, Liang et al. (2023) demonstrated the utility of mNGS in identifying multiple pathogens simultaneously by detecting coinfections with SARS-CoV-2 and influenza A [28]. As well as this, Xie et al. (2023) demonstrated that mNGS provided a sensitive and unbiased method for detecting multi-pathogenic pneumonia in HIV-infected patients [29]. These studies demonstrate mNGS’s versatility and effectiveness in diagnosing a variety of infectious diseases, providing clinicians with a comprehensive and advanced diagnostic tool.

While mNGS holds immense potential for a wide range of applications, implementing it into clinical settings for diagnosing infectious diseases poses several challenges. The challenges associated with the standardization of workflows in mNGS have a significant impact on the turnaround time and costs [30,31], data quality, reproducibility [32], comparability [33], and biological interpretation [34,35,36]. Therefore, recent studies have made significant progress in standardizing metagenomics workflows both in the wet laboratory (wet lab) [37,38] and the dry laboratory (dry lab) [39,40,41]. For instance, Parker et al. (2023) developed a sample-to-answer workflow called PanGIA that includes simplified, standardized wet lab procedures and data analysis with an easy-to-use bioinformatics tool [38]. In addition, the International Organization for Standardization (ISO) has also published international standards for the data processing of shotgun metagenomic sequences (ISO24420:2023) [42] and for the workflow and quality evaluation of sequencing data from massively parallel sequencing (ISO20397-1:2021 and ISO20397-2:2021) [43,44]. In this review, different mNGS approaches and challenges in infectious disease diagnosis will be firstly discussed, along with the strategies and validation of mNGS, and then implementing it into clinical settings will be discussed based on different published guidelines and international standards.

## 2. Understanding the Role of ISO Standards in Workflow Standardization and Different mNGS Approaches

The operation of mNGS is mainly divided into two areas, which are the bioinformatics pipeline level (dry lab) and then the entire mNGS pipeline level (wet lab). A well-established protocol for experiment design, starting from sample collection, processing, nucleic acid extraction, library preparation, and sequencing in the web lab, to binning, assembly, annotation, and visualization in the dry lab, is imperative to ensure reproducibility and accuracy of mNGS workflows. An integrated quality management approach, in line with regulatory standards, is required for any NGS-based implementation in the clinical laboratory [45]. The wet lab workflow and quality evaluation of sequencing data in the dry lab section should meet ISO20397 standards. The data processing specific to mNGS should follow the general requirement under the ISO24420 standards. To be used in clinical laboratories, assays should follow ISO15189 [46,47], a global quality standard for use in clinical laboratories, and standardized standard operating procedures (SOP). Changing any component of an assay should be validated and meet acceptable performance criteria before testing on patients, including reagents, equipment, sequencing instruments, and bioinformatics tools. Continual updating of the SOP, the reagent manual, the equipment calibration, and participation in external quality assessment (EQA) schemes are necessary to incorporate intermediate technological advancements. Besides that, laboratories must define clearly the intended clinical use for unbiased metagenomic testing, as well as the types of pathogens that will be detected and reported. There are many factors to be considered, including turnaround times, specimen requirements, library preparation protocols, sequencing platforms, sequencing depth, quality control, data analysis, costs, and clinical relevance [48]. Results review and reporting workflow should be clearly defined by laboratories, so they can choose the right data analysis solutions [48]. It is essential to be familiar with all available options, along with upcoming protocols for validation, given that many workflow components starting from the sample collection to reporting are constantly being developed and optimized.

An important challenge of metagenomic pathogen detection is the validation of an assay which cannot be declared validated until it has been applied to a wide variety of microorganisms almost endlessly. Assays that are not validated may generate incorrect results, leading to incorrect clinical interpretations and treatment decisions. There are no definitive recommendations for the clinical implementation of mNGS testing, even though the US Food and Drug Administration (FDA) has provided guidelines for clinical validation of NGS-based infectious diseases testing [49]. It is therefore recommended to learn from failed experiments and to use representative organisms to evaluate performance characteristics, followed by ongoing monitoring of assay performance and independent confirmation if unexpected results occur.

### 2.1. Understanding the Current mNGS Strategies and Challenges for Infectious Disease

A number of mNGS approaches have shown promise for clinical diagnosis of infectious diseases based on metagenomics, including short-read sequencing, long-read sequencing, and RNA sequencing.

#### 2.1.1. RNA Sequencing

In metagenomics, RNA sequencing is useful for revealing the active gene expression profiles of microorganisms, thereby revealing the functional activities of the microbes and their responses to their environments [50]. Gene expression profiles within complex microbial communities can be comprehensively analyzed using this approach, providing insights into the functional activities and responses of microorganisms [51]. A major advantage of RNA sequencing in metagenomics is its ability to capture real-time functional dynamics of microbial communities, thus allowing the monitoring of gene expression changes in various environments [51]. Additionally, RNA sequencing can also help detect novel RNA viruses and provide insight into RNA viral diversity and evolution in metagenomic samples [52]. However, RNA sequencing in metagenomics poses challenges, such as the inherent instability of RNA molecules, which results in transcript degradation and biases in transcript representation [53]. This instability can lead to inaccurate gene expression quantifications and data analysis artifacts [53,54]. Moreover, microbial RNA samples containing host-derived RNA can complicate the analysis and interpretation of metagenomic data; therefore, robust bioinformatics pipelines are required for precise taxonomic and functional analysis [54].

#### 2.1.2. Long-Read Sequencing

Long-read sequencing in metagenomics involves the usage of sequencing technologies that produce longer DNA reads, such as those provided by Pacific Biosciences and Oxford Nanopore, for analyzing the genetic contents of microbial communities present in environmental samples. This approach offers several advantages in metagenomic studies. In comparison to short-read technologies, long-read sequencing enables a more accurate reconstruction of the bacterial genome by enabling the assembly of complete bacterial genomes with a greater degree of contiguity [55]. Longer reads also enable better bridging of repetitive sequences within and between genomes, resulting in more complete genome assemblies [56]. Further, long-read sequencing improves the accuracy of metagenomic annotations and profile recovery by allowing complete genes, operons, and repetitive elements to be retrieved before assembly [57]. Despite its advantages, long-read sequencing in metagenomics also presents challenges. The higher cost of generating long reads compared to short reads may limit the adoption of long-read technologies for metagenomic studies [58]. Furthermore, long-read sequencing may suffer from high DNA requirements and sequencing errors, which can affect accurate gene prediction and assembly [56]. Moreover, long-read sequencing for metagenome assembly may be limited by limitations such as incomplete databases, insufficient read lengths, and high sequencing error rates [58].

#### 2.1.3. Short-Read Sequencing

Short-read sequencing in metagenomics typically generates DNA sequences that are relatively short, often around 50 to 300 base pairs. There are several advantages and challenges associated with this method, which is typically used by platforms like Illumina and SOLiD. Some major advantages are greater throughput, lower cost, and well-established data analysis pipelines [59], and the ability to target specific regions of interest using enrichment methods such as amplicon sequencing [60,61] or capture-based techniques [62,63]. As a result of targeted sequencing, low-abundance pathogens within complex microbial communities can be detected more sensitively, improving diagnostic capabilities [64]. In addition, short-read sequencing with enrichment can provide higher coverage depth, allowing for greater accuracy when it comes to the identification of microbial species and genes for antimicrobial resistance [65]. As an example, Illumina amplicon sequencing enables characterization of bacterial communities at a high resolution by targeting specific regions of the 16S ribosomal RNA gene, providing valuable information about the diversity and composition of bacterial communities [60,61]. Using microbiota from the gut, soil, and bacterial infections has been crucial to the study of various ecosystems, providing valuable insights into microbial populations and their interactions [60].

However, short-read sequencing can also pose challenges in metagenomics. One significant limitation is the difficulty in assembling complex metagenomes due to the short lengths of the reads, which can lead to fragmented assemblies [66]. It is especially difficult to reconstruct individual genomes in mixed microbial communities when multiple genomes are present [67]. Additionally, short-read sequencing may encounter repetitive regions and structural variations, making genome assembly less accurate and complete [68]. For the target sequencing, one key challenge is the risk of amplification bias during enrichment, which causes the data to reflect a skew of certain microbial taxa or genomic regions, which can affect microbial diversity quantification and prevent detection of rare or novel organisms [69].

## 3. Idea of Metagenomics Sequencing Assay Implementation Workflow

Implementing metagenomics sequencing assays for infectious disease diagnosis involves several key considerations. Currently, the FDA is actively assessing the components necessary for the approval of NGS-based tests for infectious diseases, which are primarily used to detect antimicrobial resistance and identify pathogens [70]. The FDA released guidelines for designing, developing, and validating next-generation sequencing tests, paving the way for routine NGS-based testing in clinical laboratories [71]. In the following sections, we will discuss how the workflow of mNGS implementation can be divided into several steps, based on various studies, guidelines, and international standards, including assay description and definition, workflow integration and optimization, assay validation, and clinical evaluation (Figure 1).

## 4. Assay Description

Assay description and definition are crucial components in the process of assay integration, as they provide a clear understanding of the assay methodology and parameters before its integration into a real system and the establishing of standardized protocols. According to the guidelines from the FDA, “Infectious Disease Next Generation Sequencing Based Diagnostic Devices: Microbial Identification and Detection of Antimicrobial Resistance and Virulence Markers”, because of the large diversity of specimen types and infectious disease agents that can be present in samples, straightforward pre-analytical, biochemical, or bioinformatics processes are not allowed. Each unique specimen type may require a different protocol for nucleic acid extraction and library preparation, as well as a unique bioinformatics algorithm to generate the final clinical result; therefore, the completed assay description needs to be defined clearly [49].

### 4.1. Deciding on Laboratory Space Organization and the Arrangement of Equipment

Since mNGS is unbiased and there is an increased risk of detection and interference caused by cross-contamination, proper laboratory organization and diagnostic workflow are essential [37]. Workflow must be unidirectional between the pre-amplification and post-amplification areas. Reagent preparation rooms must be clean, free of samples, and separated from post-amplification and pre-amplification areas to avoid any contamination of the reagent and the mastermix. A laboratory’s equipment, such as freezers, refrigerators, pipettes, safety cabinets, thermocyclers, vortexers, centrifuges, etc., should be clearly labeled and assigned to different rooms. All equipment must be calibrated by accredited bodies and the calibration intervals of all equipment should be based on the ISO15189 standards.

### 4.2. Defining the Scope of Pathogens

The laboratory should first define the pathogens that will be detected by the assay, based on the intended use, the type of specimen, and the characteristics of the patient. The choice of pathogens influences the choice of extraction method, sequencing strategies, target enrichment or host nucleic acid depletion, reference database, sequencing depth and coverage, and data analysis tools. Besides that, there are several factors to consider when determining the scope of pathogens. These factors include genotype characteristics, environmental conditions, and non-random factors that influence pathogen damage to varieties [72]. Additionally, the transmission dynamics of pathogens can be influenced by challenges encountered in zoonotic disease surveillance, insufficient spatial information, and a lack of understanding of surveillance scope [73]. Emerging infectious diseases are significantly correlated with socioeconomic, environmental, and ecological factors, providing a way to identify regions that are likely to develop new diseases [74]. An integrated approach to zoonotic spillover requires an alignment of ecological, epidemiological, and behavioral factors influencing pathogen exposure, as well as factors influencing susceptibility to infection within humans [75]. Furthermore, the vector-to-host ratio plays a pivotal role in determining the severity of pathogen transmission and infection risk [76]. Moreover, microbial pathogens’ vulnerability affects their resistance and pathogenicity, making them crucial in determining the scope of pathogens [77]. Factors such as human connectivity, increasing antimicrobial resistance, and dynamic human behavior influence prevention and control, affecting the scope of pathogen dynamics [78].

### 4.3. Defining the mNGS Strategies

mNGS strategies should be determined according to the application before implementing mNGS assays. To improve the detection and analysis of microbial DNA in complex samples, mNGS strategies primarily use sensitivity and enrichment or depletion methods. mNGS has a major limitation with the presence of a high background, either from the host or from the microbiome. As an example, Wilson et al. evaluated the clinical effect of mNGS in diagnosing meningitis and encephalitis, highlighting the limitations associated with low sensitivity when the human host or microbiome background is high [25]. Furthermore, Salipante et al. were able to show the limitations of low sensitivity due to high background from the microbiome when they compared the performance of Illumina and Ion Torrent sequencing platforms for bacterial community profiling [79]. Additionally, the importance of monitoring low-biomass samples for background contamination was also highlighted by Weyrich and colleagues [80].

To efficiently perform mNGS and improve microbial DNA detection in different samples, a host depletion approach is crucial. A variety of approaches have been used to overcome host contamination, including capture probes for subtractive hybridization [21,81], CRISPR-Cas9 cleavage on target sequence [82] and ribonuclease (RNase) H-based depletion methods [83], as well as nanopore adaptive sequencing [84]. In the case of RNA libraries, DNase I treatment should be performed after extraction to remove residual human background DNA [85]. Aja-Macaya et al. have recently developed an efficient method for enriching the mNGS of monkeypox viruses using saponin/NaCl combination DNase treatments, combined with high g-force centrifugation, resulting in 96% of reads being classified as not human DNA using the enrichment method, and 5–10% using the non-enrichment method [86].

### 4.4. Defining the Test Methodology

A complex and computationally intensive process is involved in transforming raw sequence data into actionable information in NGS, starting with sample collection, extraction, fragmentation, library preparation, sequencing, data analysis, and interpretation of the results (Figure 2). As a brief overview, the DNA/RNA are first extracted from the samples. Following the extraction of DNA/RNA, a library is constructed where the DNA of each organism is sheared into fragments and inserted into adapters containing barcodes to allow multiplexing of hundreds of samples based on the quality criteria for the selected NGS platforms and applications. Using a preferred NGS technology, these individual libraries are pooled together and submitted for analysis. After the sequencing has been completed, several bioinformatics steps need to be performed, including quality control, read alignment or assembly, variant calling, and annotation, which is dependent on the specific application [87]. An analysis of the assembled genome against reference strains facilitates the identification of pathogens, strain typing at high resolution, and prediction of important phenotypic characteristics (e.g., virulence, resistance to antibiotics) [88]. There is a need for well-curated and up-to-date reference databases since microorganisms are constantly evolving and plasmids encoding traits of virulence and antimicrobial resistance can be exchanged across strains and species. By comparing assembled genomes with others, we can look for evidence of transmission based on phylogenetic clustering [88]. Different bioinformatics tools are required for each step, including assembly, strain typing, phenotyping, and clustering. Therefore, bioinformatics expertise and computational resources are required to perform the analysis using various software tools and pipelines [87,88].

### 4.5. Defining the Possible Risk/Errors in the Whole Workflow

#### 5M1E Methodology

The 5M1E methodology (manpower, method, machine, material, measurement, and environment) was mainly involved in the process of the root cause analysis to enhance problem-solving capabilities, optimize procedures, and maintain high-quality assurance standards. A systematic approach, the Plan–Do–Check–Act (PDCA) cycle, was also incorporated into the 5M1E methodology, which involves four stages: planning, implementing, evaluating, and action-taking based on the evaluation. Therefore, the 5M1E methodology needs to be applied to the whole process, starting from pre-implementation to implementation steps, including assay design, quality control matrix determination, and assay validation. An overview of the subject, action, and objective based on the 5M1E methodology using an NGS assay is shown in Table 1. The 5M1E methodology can also be applied in molecular laboratories. The potential reasons for errors are listed in Table 2.

### 4.6. Defining the Quality Control Metrics

Quality control is a critical aspect of mNGS to ensure the accuracy and reliability of the data. It is necessary to establish quality control measures and to apply them to every run of the assay. Specimen acceptability, nucleic acid quantification, and library qualification and quantification should be monitored since libraries with inadequate fragment sizes and underloading can result in reduced coverage [124]. A quality control (QC) process should be carried out on raw mNGS data to ensure that the data are clean. The QC metrics and follow-up actions are summarized in Table 3 following ISO24420:2003, ISO20397-1:2022, Clause 4 and 8.3, ISO20397-2:2021, 4.3, and different metagenomics studies.

## 5. Assay Integration and Optimization

After defining the assay and describing its components, the next crucial step in the workflow is the integration and optimization of the entire process. This involves streamlining the various stages of the assay, from sample collection to data analysis, to ensure efficiency and accuracy.

### 5.1. Choice of Sample Type

The presence of low pathogen abundance is one of the challenges in detecting and identifying pathogens in complex microbiomes [141]. Microorganisms that live in the environment or serve as commensals may make it more difficult to identify true pathogens [141]. Therefore, it is important to select the right sample type based on the type of microorganisms to be studied. Multiple factors should be considered when determining the appropriate sample type for metagenomic sequencing. There are several factors to consider, including the research objectives, the target microbes, and the resources and technologies available. For instance, if viral genomes are of interest, blood or respiratory samples may be appropriate [142]. In contrast, for bacterial infections, samples from the lower respiratory tract or clinical specimens may be more suitable [143]. Data from mNGS should also be used for the intended purpose. In outbreak investigations, the source of infection and its spread can be determined by sample collection from affected individuals or environmental sources [15,144]. Similarly, contaminated food samples or samples from affected individuals can be tested for pathogens in foodborne outbreak investigations [144]. The use of mNGS for AMR surveillance can be carried out with samples from sewage or wastewater [145], fecal samples [146,147], or respiratory samples, such as nasopharyngeal swabs or sputum samples from human populations. It is possible to obtain valuable information from these types of samples regarding how antimicrobial resistance genes are prevalent, diverse, and transmitted within populations. For instance, fecal samples can provide insight into the gut microbiota and its role in the development and transmission of AMR [147]. The gut microbiota has been altered in patients with COVID-19, and the abundance of specific species of bacteria has been correlated with the severity of the disease [147].

### 5.2. Sample Preparation

Preparation of samples is the first and most important step in any metagenomics analysis. Ideally, metagenomic sampling aims to collect enough biomass to perform sequencing while minimizing contamination. Samples from the same environment can have varying microbial content, making it more difficult to detect statistically significant and biologically relevant differences between them. To ensure sufficient statistical power, it is crucial to design experiments appropriately, especially for small samples [148]. The conditions under which samples are collected and preserved can affect the quality and accuracy of metagenomic data, and they are also important confounders when analyzing data from different studies [148]. In addition, optimizing processing conditions for all types of samples is extremely important, and methods validated for one type of sample cannot automatically be adapted for other types [148]. Besides that, it is important to keep track of and consider conditions such as the length of time between sampling, freezing, and thawing that can affect the microbial community during the experiment [148].

### 5.3. DNA/RNA Extraction

The DNA/RNA extraction methodology can influence downstream sequencing data. Bead-based and column-based are the two main nucleic acid extraction approaches. For infective disease diagnosis, nucleic acid extraction methods should be selected based on their advantages and limitations, as well as the specific diagnostic requirements. Compared with conventional liquid-phase extraction methods, the bead-based extraction method is more suitable for mNGS in the context of infectious disease diagnosis because of the higher yield, purity, and reproducibility. According to McEvoy et al. (2020), the use of solid-phase reversible immobilization beads (SPRI) in DNA purification is favored by many protocols because of its ease of use, cost-effectiveness, size selectability, and automation capability [149]. Among the many methods of extracting nucleic acids from host samples, bead-based extraction was shown to be the best suited for depletion of host nucleic acids for diagnostic metagenomics of infectious diseases [150]. In diagnostic settings, column-based methods are demonstrated to be versatile and effective in detecting pathogens such as herpes simplex virus and parvovirus B19 [151,152]. However, column-based methods may result in DNA loss and lower recovery rates, highlighting the superiority of bead-based extraction for mNGS in capturing the comprehensive and high-quality metagenomic profile essential for accurate infectious disease diagnosis [153].

As mentioned before, each sample type should have unique extraction methods according to the guidelines from the FDA. Due to the differences in the composition of cell walls among plants, animals, bacteria, and fungi, the extraction of DNA and RNA for mNGS requires distinct and tailored methods. Plants with rigid cellulose-based cell walls may require methods like cetyltrimethylammonium bromide (CTAB) or using a plant-specific kit, such as the DNeasy Plant Mini Kit from Qiagen [154], while animals lacking cell walls but containing other cellular components may benefit from phenol-chloroform extraction and a column-based purification kit, such as the DNeasy Blood & Tissue Kit from Qiagen. Bacteria, with unique cell wall structures in Gram-positive and Gram-negative species, may necessitate enzymatic lysis or bead beating methods to disrupt the cell wall and membrane, in order to extract DNA and RNA [155]. Fungi with chitin-based cell walls may require enzymatic digestion for efficient nucleic acid extraction [156].

The choice of lysis approaches is also the main factor to affect the downstream sequencing result. For example, some cells are less susceptible to lysis techniques, resulting in the under-representation of DNA from these organisms [157,158,159,160]. Therefore, the extraction method must be effective for different microbial species to prevent easy-to-lyse bacteria from dominating the sequencing results. A variety of lysis approaches can be used to extract DNA, including mechanical and chemical methods. Using mechanical methods, such as bead beating, can result in DNA fragmentation and loss during library preparation methods involving fragment size selection. Hence, it is necessary to use the appropriate extraction technique, to ensure that a sample is representative of all cells present with high quality and sufficient quantity. On the other hand, contamination occurring during the sample extraction and library preparation also can affect the quality of the downstream sequencing data. There are several sources of contamination, including the environment, personnel, machines, as well as laboratory reagents [161]. It is particularly problematic for samples with low biomass, as there is less signal to compete with low contamination levels [91]. As a precaution, ultraclean reagents are recommended and no template sequencing controls are included [91]. In RNA extraction, to get better yields from in vitro transcription reactions, degradation during RNA purification protocols, and lower yields from RNA extraction, decontamination with the RNase decontamination solution is crucial. It is also possible to obtain RNA from total nucleic acid by treating it with DNase. Carrier RNA should not be used for RNA mNGS, since it will be sequenced along with the sample RNA and affect the viral sequencing depth [162].

### 5.4. Fragmentation

Before library preparation, some sequencing methods, such as short-read sequencing, require fragmenting the template DNA, cDNA, or RNA. As a result of fragmentation, DNA or RNA can be made into a size range suitable for the particular method and sequencing platform. It can be either mechanical or enzymatic fragmentation. Long RNA fragments are typically fragmented chemically. It is important to consider how fragmentation methods impact library coverage, to avoid GC biases introduced by specific fragmentation methods. Consideration should also be given to the amount of starting material and the potential sample loss associated with each method. In mNGS, fragment size can affect sequencing depth and the ability to assemble complete genomes from data [163]. For instance, small microbial genome size and limited fragment size necessitate a higher number of fragments for metagenomic sequencing [163]. Nevertheless, mNGS presents a substantial challenge due to the complexity of the data, which may consist of sequence fragments from many different genomes [164]. Due to the unassembled nature of most mNGS, conventional methods of gene discovery are not relevant [165]. To address this issue, metagenomic approaches involve the sequencing of randomly sheared fragments, known as shotgun sequencing, to provide a potentially more accurate characterization of microbial diversity [166].

### 5.5. Library Preparation

The choice of library preparation method can significantly influence downstream sequencing quality issues, and choosing the right library preparation methods for different sample inputs is crucial for ensuring robust data quality across samples in different preservation conditions [128]. Sequencing bias in library preparation kits is important in mNGS, which involves a wide variety of microbes with varying GC contents [135,167]. Additionally, some studies have also demonstrated that library preparation procedure and sample characteristics influence metagenomic profile accuracy, which is important in choosing the best preparation method for metagenomic libraries based on sample characteristics, including community types, compositions, and DNA amounts [168,169]. According to Wang et al. (2022), different sequencing library preparation methods were evaluated to get efficient and high-fidelity metagenomic data from samples with low microbial biomass, emphasizing how library preparation is crucial for overcoming challenges associated with low biomass samples [170]. Similarly, Jones et al. (2015) found significant differences in taxonomy between next-generation sequencing library preparations, highlighting the significance of library preparation methodology for predicting genomic and functional outcomes [171].

### 5.6. Sequencing Platform

When selecting a sequencing platform for integrating assays in mNGS, it is crucial to consider various factors, such as cost, the amount of sample input, and required concentration and fragmentation size ranges. The sequencing depth required often influences the choice of platform, as different platforms come with distinct advantages and challenges [172,173]. Third-generation sequencing platforms, such as Oxford Nanopore Technologies (ONT), have demonstrated potential in offering precise species resolution compared to Illumina sequencing [174,175]. Parameters like read length, sequencing coverage, and sequencing errors play a significant role in the efficiency and accuracy of metagenomic sequence assembly [42,176]. Therefore, a thoughtful evaluation of the sequencing platform is necessary to ensure optimal performance and dependable outcomes in metagenomic studies.

#### Bioinformatics Analysis for Metagenomics

For metagenomic data analysis, it is often difficult to determine which sequences belong to a given pathogen in mNGS datasets, as the majority of sequences come from the host, and only a very small fraction of sequences come from the pathogen. Therefore, according to ISO24420:2023 Clause 5.2.3, it is recommended that host reads from all organisms be removed by mapping them to a genome reference such as UniRef or the Unified Human Gastrointestinal Protein Catalog [42,128]. mNGS involves a series of bioinformatics steps to analyze raw data effectively, including preprocessing of sequencing reads, de novo metagenome assembly, genome binning, taxonomic and functional analysis of genomes, fragment recruitment to reference genomes, and metagenomic assembly and analysis (Table 4). In dealing with metagenomic datasets, there are three main approaches, including marker gene analysis, binning, and assembly [115,165]. 

The marker gene analysis compares each sequence with a database of taxonomically and phylogenetically informative sequences called marker genes. The sequences are then taxonomically annotated after the similarities are assessed [165]. According to ISO24420:2003, for obtaining a higher-level taxonomy profile (e.g., species, genus, order, or phylum), including metagenomic linkage groups (MLGs), metagenomic clusters (MGCs), and metagenomic species (MGS), the best taxonomy profile method should be selected based on the data and the application needs [128]. The concept of binning refers to the clustering of sequences according to taxonomic groups, such as species, genus, or higher taxa [165]. Assembly is to combine the small sequences from your sample to make longer sequences represent genomes [165]. In the absence of a reference genome dataset, such as soil or ocean samples, sequence assembly should be used [128]. In de novo assemblies, contigs or scaffolds are derived without any reference from sequence fragments [128]. The choice of sequence assembly software should take into account the extent to which accuracy, contig size, input data type, and computational resources are important [128]. Therefore, different profiles, such as phylogenetics/taxonomic abundance and contigs/genome coverage, are able to provide valuable indications for function annotation and taxonomic classification. 

### 5.7. The Choice of Reference Database

Additionally, one of the limitations of metagenomics is the quality and availability of reference databases. The choice of a reference database has a significant impact on taxonomic classification results [195]. Reference databases should contain all relevant genome sequences covering the entire genetic diversity of organisms and ensure that no artificial, low-quality, or incorrectly named genome sequences are included [196]. The incorrect assignment of ambiguously mapped/aligned reads or k-mers in poorly curated databases may lead to false positive results [196]. False negative results can be caused by insufficient databases that do not include newly discovered viral strains or uncommon strains [197]. In pathogen detection, compressing a database by removing duplicate sequences will reduce performance, but it is an effective way of saving storage space [198]. Generally, larger databases allow for better sub-typing/classification at the isolate level [196].

The National Center for Biotechnology Information (NCBI) is a popular database for metagenomic analysis [199]. However, a biased collection of isolated viruses may be present in the NCBI database [200]. Hence, researchers have manually identified metagenomic viral contigs to supplement the viral protein families found in the NCBI [200]. Through this approach, viral genome diversity can be expanded and viral communities can be better represented in metagenomic studies. The FDA-ARGOS database and the FDA Reference Viral Database (RVDB) are examples of databases that have been developed for diagnostic use and regulatory science [201], and which provide quality-controlled reference genomes for various pathogens. For instance, by using the MEGARes V3.0 database, antimicrobial resistance genes can be identified from mNGS data [118], while the Microba Community Profiler can be used to identify taxonomic groups from metagenomic datasets [202]. According to ISO24420:2023, taxonomy profiling should be conducted using reference datasets, such as the RefSeq complete genomes (RefSeq CG) for microbes, and the BLAST database for high-quality nucleotide and protein sequences [42]. The other databases, such as ICTV Virus Metadata Resource (VMR) (VMR_MSL38_v2, https://ictv.global/vmr (accessed on 20 January 2024), Reference Viral DataBase (RVDB) (v28.0, 22 November 2023), Reference Viral DataBase (RVDB, protein version 28, https://rvdb-prot.pasteur.fr/ (accessed on 20 January 2024), SIB Viral reference sequences, UniProt Virus proteomes, VirMet (v1.2), Virosaurus, and Virus Pathogen Resource (ViPR), are available for pathogen/viral mNGS [196]. There is also a list of simulated datasets and synthetic datasets publicly available for analysis of this type [203]. However, there may be differences between these databases when it comes to availability and coverage, and it is important to standardize data quality and representation [172]. Since reference genome databases for microorganisms are constantly updated, laboratories must pay attention to the exact versions being used as well as possible mis-annotations in reference databases [30]. A metagenomic profile can be annotated at various levels according to the reference annotation, e.g., species, genus, or higher [42].

### 5.8. Reporting

Several key findings should be reported to doctors as part of the metagenomic analysis. Firstly, the methods, limitations, and quality, including the depth of coverage of the assay, must be reported. Secondly, it is essential to report whether specific pathogens are present in the sample since this information can be used to guide treatment decisions [25]. Additionally, comparing abundances of microorganisms can help assess microbiome composition and dysbiosis [204]. Furthermore, antibiotic resistance genes and virulence factors should be reported as part of the analysis, as this information may inform treatment strategies [205]. A functional analysis should also include information from the microbiome, such as metabolic pathways or genes that play a key role in the microbiome [206]. Finally, clinical interpretation of results should be conveyed concisely and clearly, which will enable accurate diagnosis, guide treatment decisions, optimize antimicrobial therapy, and lead to improved patient outcomes [206]. For instance, the results of mNGS can give patients and clinicians confidence that unnecessary empirical treatments can be stopped and can tell them when coinfections should be ruled out, and when infectious syndromes should be diagnosed [25]. Furthermore, it may be possible to expedite or defer the escalation of empirical therapy based on mNGS results, resulting in a more timely and effective treatment for patients [207]. Through mNGS, pathogens and antimicrobial resistance determinants can also be identified. This helps tailor therapy to the specific infectious agent and potentially reduces antimicrobial resistance selection [208]. The presence of viruses of unknown pathogenicity or seldom-detected viruses may not have caused a specific disease in the past, but at a later stage could lead to a specific syndrome, which is why it is recommended to have these viruses reported [196]. An interpretation of an unknown metagenomic finding can be discussed subsequently with the clinician or noted in the report as well.

## 6. Assay Validation

According to ISO2037-1:2022 Clause 6, before accumulating validation data, a validation protocol for intended use shall be developed, implemented, and documented [43]. Through the validation process, potential failures should be identified systematically. It is also important to validate assays using samples of the type intended for use in the test, so that the test results are representative of the larger sample population [43]. Validation can be carried out with known positive and spiked specimens throughout the workflow, including pathogens that are exclusive to a particular region and those that are common colonizers, as well as microorganisms that are found as contaminants in the environment at high levels [209]. In each matrix, the limits of detection (LODs) for reference microorganisms should be determined [209]. It is important to keep in mind that sequencing depth can affect the LOD; therefore, it is necessary to provide similar sequencing depths for all samples, which can fall into an acceptance range of sequencing depths [209]. Through repeated measurements of samples with low concentrations of analytes, the limit of quantity (LOQ) can also be used to evaluate the sensitivity and reliability of an analytical method. Besides that, for tNGS, gene- and disease-specific aspects must be included in test validations for disease-specific targeted gene panels. To ensure that the most common causes of disease are detectable, common pathogenic variants must be included in the validation set, even though sequence-specific context can affect its detection [210]. Additionally, negative controls are essential to eliminate possible contamination, which can occur at any stage from sampling to sequencing. Sample blanks, nucleic acid extraction blanks, and/or no-template controls can be used as negative controls.

Considering the complexity of NGS, thorough investigation and root cause analysis are essential for ensuring assay quality, safety, and efficacy; therefore, performance validation and verification are important steps in bioinformatics analysis for mNGS. Through error-based design and optimization, potential failure points can be identified, and the level of validation and quality control can be determined. As part of “dry lab” validation, validation of the metagenomic bioinformatics software pipeline is essential. The pipeline should be validated before being used for analytical purposes, including all tools, code, environment, and network connections [42]. The in-house bioinformatics pipelines were compared with popular metagenomics pipelines using simulated in silico datasets and published raw patient datasets [211]. On the other hand, wet lab validation ensures the accuracy and reliability of mNGS results as it involves experimentally verifying the sequencing results with various laboratory techniques and reagents based on ISO15189 and ISO20397 [46,211]. In both dry and wet labs, mNGS validation should be based on the 5M1E methodology to enhance problem-solving capabilities, optimize procedures, and maintain high-quality assurance standards.

### 6.1. Reference Materials

The problem is exacerbated by the lack of reference standards in many clinical scenarios [25]. For instance, the FDA advises that when comparing results with non-reference standards, specificity and sensitivity should be expressed as negative and positive percent agreement, respectively, according to their statistical guidance on reporting results from studies evaluating diagnostic tests [212]. If no virus-positive residual clinical material is available, then mock samples can be manufactured using reference material or purchased commercially [213]. For instance, the National Institute for Standards and Technology (NIST) is developing whole-genome microbial reference standards [43]. Through genome assembly, base-level analysis, genomic purity, and genomic stability evaluations, four clinically important microbial species are being characterized by NIST [43]. To validate a microbial NGS-based diagnostic test, these reference materials (RM 8375, a 4 bacteria panel, and RM 8376, a 19 bacteria + 1 human panel) can be used in appropriate steps [43], or the reference controls also can be obtained from UK-NIBSC, the European virus archive, or the ATCC [37].

As part of accuracy studies, the choice of samples is very important. A reference control material should contain multiple viruses that have different relative loads so that specificity can be determined, or separately prepared mixtures of viral strains with different viral concentrations so that sensitivity can be assessed [37]. An evaluation of a sample analyzed with a previously analyzed or another platform, such as Sanger sequencing, should be conducted as part of mNGS validation [210]. Typically, residual samples are spiked at levels exceeding the sensitivity of the test (e.g., 10 times the LOD) after evaluation [48]. However, it may not be practical to establish clinically and analytically relevant LODs for all potential targeted organisms due to the complexity of metagenomic tests [48]. Representative and positive patient specimens for the relevant specimen type can be used as spiking sources [48,209]. By doing this, one can measure background matrix effects on a larger number of independent specimens than can be done using test-positive specimens [48]. In this step, the test needs to demonstrate its analytical performance for different microorganisms so that maximum confidence can be placed in its ability to distinguish targeted microorganisms from common microorganisms.

### 6.2. Proficiency Testing and External Quality Assessment (EQA)

To ensure the consistency and reliability of the sequencing process, quality control measures are implemented. For example, it is crucial to assess sample quality before sequencing, including DNA/RNA integrity, concentration, and potential contaminants. During library preparation, the right protocol and strategies need to be selected to minimize bias and ensure representative sequencing of the microbial community [31]. Besides that, it is important to participate in EQA programs which compare the laboratory’s performance with other laboratories to ensure accuracy. The following qualitative characteristics may be addressed by EQAs of mNGS methods for viral pathogen detection: (i) accurate identification of pathogen species, (ii) quantitative parameters, and (iii) logistic performance (turnaround time) [37]. In addition to real sequencing data, the College of American Pathologists recommends that proficiency testing be combined with silico-generated sequences or modified sequences [48]. This enables the examination of algorithms and sequence databases in greater depth. In silico specimen generation and critical data analysis steps can be performed with fewer resources because a wide variety of specimens can be generated [48,209]. In some cases, clinical samples were used to verify the conclusions drawn from in silico and contrived samples [209]. During data analysis, a minimum threshold should be established based on the diagnostic approach [210]. To determine data quality, additional metrics might include the percentage of reads aligned to the human genome, the percentage of unique reads (before duplicates are removed), the proportion of bases matching the target sequences, the uniformity of coverage, and the proportion of bases uncovered [210].

### 6.3. Enhancement of Data Analysis

Typically, NGS platforms generate complex data that need to be handled and processed in multiple stages. Due to the use of separate tools for data analysis, independent of wet laboratory procedures, and the likelihood that these tools may be customized in the laboratory, a separate validation process for the analytical pipeline should be conducted during the initial test development stage. After that, the pre-validated pipeline can be used to validate both the wet laboratory and the analytical pipeline within the end-to-end test validation. The laboratory should also document any validation data provided by vendors when using commercially developed software, but an independent validation should also be performed [210]. Therefore, it is necessary to determine the parameters and thresholds for determining whether the overall sequencing run is successful, such as the number of reads aligned, the number of duplicates, the average coverage depth, and the range of insert sizes.

Meanwhile, identifying the threshold for coverage is very important to ensure that sufficient coverage and allelic fraction are achieved for variant calling and the sensitivity and specificity of analytical tests. The success of mNGS is influenced by various factors, including genetic diversity, sequencing depth, and coverage. Metagenomic contigs are normally categorized at the species level based on their sequencing depth and coverage [214]. Additionally, algorithms for estimating coverage in metagenomic datasets can help inform decision-making about which assembling approach to use, especially with uneven distributions of coverage depth [215]. The extent of coverage is important for recovering microbiome information, and it directly affects metagenomic processing fidelity [216]. Furthermore, metagenomics coverage theory emphasizes the importance of adequate coverage for meaningful results, defining what depth of sequencing is needed within an experiment [217]. Further, it is important to understand the microbial ecology and gene diversity within diverse environments for the impact of sequencing depth on the characterization of the microbiome and resistome [218]. Metagenomic datasets can be interpreted in a straightforward manner based on robust statistical frameworks, which can be used to assess the completeness of mNGS by determining how much of the genome is covered by sequencing reads [219]. It is also crucial to estimate the metagenome coverage to ensure high sensitivity in the detection of antimicrobial resistance genes or other functional elements [220]. As mentioned before, metagenomic analyses are directly impacted by sequencing coverage depth. As a guideline for best practices in shotgun metagenomics, 1 Gb of sequencing depth per sample should be used [221]. Nevertheless, mNGS coverage requirements can vary depending on the complexity of the microbiota and the specific research objectives. To successfully sequence metagenomic samples, a minimum coverage depth is needed to identify minor variations within viral quasispecies [8]. Additionally, a 15X genome coverage limit has been determined for an assembly-based method of detecting antimicrobial resistance genes [220], which highlights the importance of detecting specific genetic elements within metagenomes with adequate coverage.

### 6.4. Performance Characteristics

During the test validation, performance characteristics such as precision reproducibility and repeatability, analytical specificity and false positive rate, analytical sensitivity and false negative rate, and LODs should be determined to provide enough evidence on the accuracy of the test. The final analytical parameters must reflect the entire testing process.

#### 6.4.1. Precision Reproducibility and Repeatability

Various instruments, reagents, and techniques can introduce random imprecision at each step in multistep and complicated procedures. To minimize variation, instruments, reagents, and personnel must be qualified and verified. Assay validation should quantify variations that can occur. Because there are many sources of variation, it is not practical to evaluate each source separately. To ensure quality, it is recommended to assess a minimum of 3 to 20 samples with known concentration or series dilution across all steps and over a long period, including all instruments, testing personnel, and multiple lots of reagents [124,209]. Quantitative precision testing should be performed both within runs (repeatability) and within laboratories (reproducibility) by using defined LOD samples. Before acquiring validation data, acceptance criteria must be established as well. Duplicate analyses of the same sample should be performed to verify reproducibility and calculate the coefficient of variation (CV%). In addition, results from different laboratories can be compared to evaluate interlaboratory performance. A laboratory’s intermediate precision is the degree to which results of tests performed on the same test items over a long period are closely aligned, considering differences in laboratory conditions such as different operators, equipment, or days [222]. Analysis and comparison of sequencing data should be carried out throughout the validation process using appropriate bioinformatics tools as well [223]. Besides that, a PCR-based approach can be used to verify the quantitative accuracy [209].

#### 6.4.2. Analytic Sensitivity and Analytical Specificity 

Analytic sensitivity refers to how accurately the mNGS assay can detect and identify target microorganisms [48]. The analytical sensitivity of mNGS refers to the method’s ability to detect and identify pathogens accurately, directly impacting its reliability. False negative rates are the rate at which true positive results are incorrectly classified as negative results. Both parameters are critical for determining metagenomic sequencing’s effectiveness in diagnosing infectious diseases. Several variables determine the sensitivity and specificity of pathogen detection, such as nucleic acid extraction efficiency, size of the pathogen genome, quality of library preparation, number of sequencing reads derived from a given specimen (coverage), the composition of samples, background noise sequences, reference sequence availability, sequencing depth, the precision of classification algorithms, and the confidence required to identify pathogens [48]. To maximize the sensitivity of mNGS assays, it is critical to extract nucleic acids efficiently and prepare libraries in a quality manner, because the number of pathogens may change the total nucleic acid yield, resulting in a sequence library containing both patient nucleic acid as well as microbial nucleic acid. There is evidence that mNGS can enhance pathogen detection in clinical samples, especially when low pathogen titers or high nucleic acid backgrounds may yield false negative results [30]. For example, mNGS has a high false negative rate when identifying pathogens such as *Streptococcus agalactiae* in infectious native valve endocarditis [212]. Hence, it is imperative to interpret negative results carefully, especially in samples that have a high host background, since they are at higher risk of false negative results [25]. There are other protocols available for enriching pathogen nucleic acids [81,224], depleting host nucleic acids [225], or removing part of sequencing libraries [82] to increase sensitivity. 

Analytic specificity in the context of mNGS refers to the ability of the sequencing method to accurately identify and distinguish specific microbial species or genetic elements within a complex sample [204], while the false positive rate represents the percentage of negative results that are incorrectly identified as positive [226]. The analytical specificity of clinical metagenomic testing can be influenced by several factors that are not taken into account in traditional validation guidelines. These include DNA fragments contaminating reagents and consumable surfaces from a variety of microorganisms, genome sequence similarity among microorganisms, incorrect reference genome sequences, and the difference between clinical isolates and reference strains. Limited analytic specificity can occur as a result of the misclassification of microorganisms or nucleic acids or the contamination of reagents, which can be attributed to algorithms used or to reference databases [48]. The identification and mitigation of problems can be assisted by extensive in silico validation with particular emphasis on microbes with sequence homologies with relevant pathogens [48].

#### 6.4.3. Limit of Detection

The LOD refers to the lowest concentration or abundance of pathogens or genetic material that can reliably be detected and quantified in mNGS. It is crucial to assess the accuracy and sensitivity of mNGS in identifying pathogens in clinical samples based on this parameter. For instance, Greninger et al. (2010) demonstrated that mNGS can detect pathogens near the limits of detection for RT-PCR assays even at low concentrations. This illustrates what mNGS can achieve at low concentrations, allowing pathogens that would be difficult to identify using traditional techniques to be detected [227]. By incorporating computational approaches and modeling techniques, LODs for mNGS have been refined. Serial dilutions of a clinical sample or external controls with a known, quantifiable pathogen can be analyzed to determine the LOD, or a set of calibrated internal controls can be used. For example, receiver operating characteristic (ROC) curves have been used to refine similarity searches, and a reliable, fixed bitscore value has been calculated across the sequence of the target gene to maximize sensitivity and specificity in the detection of short-gene fragments within metagenomes [228]. To determine the LOD, cut-off thresholds are defined for coverage and sequence depth that are used in decision-making in the context of sample composition, since the host nucleic acid burden can rapidly alter this LOD [37].

## 7. Future Perspectives and Challenges for Implementing mNGS in Infectious Disease Diagnosis

The future of mNGS holds a lot of promise and potential applications in a range of fields. It has the potential to provide significant new insights into the pathogenesis of new and emerging infectious diseases, as well as immunopathological phenomena relating to host–pathogen interactions [229]. In the future, deeper insights into genetic responses, mechanisms, and genetic variation may be possible due to the technology’s ability to interrogate increasingly complex microbial communities [230]. However, several challenges need to be considered for successful integration (Table 5).

### 7.1. Turnaround Time and Costs

NGS can be costly, requiring specialized equipment and expertise. Clinical settings with limited resources may be unable to afford sequencing and high-performance computing infrastructure [30]. In the clinical setting, providing timely results from mNGS is crucial to rapid diagnosis in clinical settings and is also challenging. Data analysis pipelines and sequencing technologies need to be improved to reduce turnaround time [25]. Future perspectives involve enhancing the cost-effectiveness of NGS platforms and optimizing bioinformatics pipelines to achieve cost reduction. For example, the RAPIDprep assay was introduced as a simple and fast protocol for the RNA mNGS of clinical samples which provides a cause-agnostic laboratory diagnosis of infection within 24 h of sample collection by sequencing ribosomal RNA-depleted total RNA [31].

### 7.2. Standardization and Quality Control

The standardization and quality control of mNGS are crucial for ensuring data accuracy and comparability. Standardizing laboratory procedures to ensure the quality and interoperability of Big Data generated by sequencing remains a crucial issue [34]. In 2011, Field et al. noted the Genomic Standards Consortium (GSC) as driving community-based standardization activities that aimed at improving the quality and quantity of contextual data associated with public collections of genomes and metagenomes [241]. The GSC has set out a minimum level of information for a metagenomic sample, emphasizing the importance of standardizing the context [242]. However, it is challenging to standardize mNGS because there are no universally accepted protocols and standards. The development of consensus standards should be the focus of future efforts through collaboration between researchers, industry stakeholders, and regulatory bodies. The process may involve the formation of international working groups or consortiums that set standards for sample preparation, library preparation, sequencing, and bioinformatics analysis [35]. Automated technologies can also increase reproducibility and reduce human-induced variability in metagenomics sequencing workflows [243]. For instance, PacBio collaborated with an automation partner to create fully automated protocols to prepare samples for sequencing [244]. Besides that, it is important to launch proficiency testing programs for mNGS to help the standardization and implementation of mNGS [233]. With the bioinformatics field continuing to evolve, efforts should be focused on the establishment of standardized data formats, ontologies, and metadata definitions for metagenomic sequencing data. In this way, data sharing will be facilitated, and integration and meta-analysis will be improved, ultimately resulting in better reproducibility and comparability of results [141,236,243,244]. The standardization and quality control of mNGS will be enhanced by future technological advancements through collaboration and open sharing, including novel sequencing platforms, improved bioinformatics tools, and innovative quality control assays [237,238].

### 7.3. Bioinformatics and Data Analysis

Bioinformatics tools and sequencing technologies will improve the sensitivity, specificity, and speed of sequencing. Validation studies and standardization efforts will establish WGS and mNGS as an effective and reliable diagnostic method [141]. The use of WGS can improve patient outcomes and guide targeted therapies by helping to identify genetic variants associated with treatment response, disease susceptibility, and adverse drug reactions [239]. Technological advancements, such as nanopore sequencing, will enable point-of-care sequencing and expand the accessibility of WGS in a variety of clinical settings while improving its speed, accuracy, and cost-effectiveness [240]. Ultimately, mNGS and WGS are set to transform research, diagnostics, and environmental monitoring with ongoing advancements and applications across diverse fields. Therefore, the development of secure and scalable data storage solutions and user-friendly software and databases to identify pathogens, predict antimicrobial resistance, and analyze outbreaks makes these tools more accessible to healthcare professionals [236,237].

## 8. Conclusions

In conclusion, the strategies and challenges of implementing mNGS are discussed in this review. The validation under different ISO standards, as discussed in this review, is important to minimize any errors in the workflow and to ensure the accuracy and reliability of NGS in infectious disease diagnosis. As the field of clinical mNGS continues to evolve, standard protocols, bioinformatics pipelines, and reference databases are essential for addressing the challenges and facilitating the wide adoption of mNGS in clinical microbiology laboratories. In the future, the development of reference standards and guidelines will be crucial to ensuring the accuracy, reproducibility, and reliability of mNGS.

## Figures and Tables

**Figure 1 ijms-25-03333-f001:**
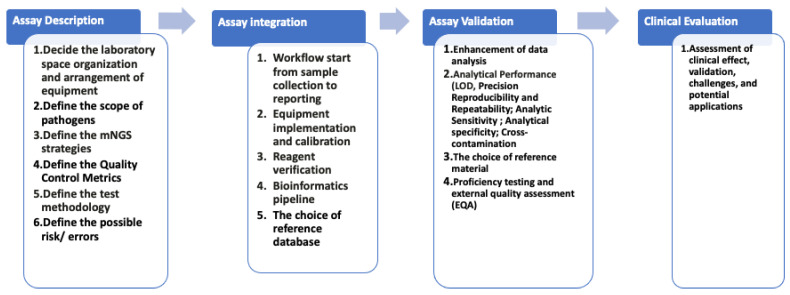
Overview of metagenomics sequencing assay implementation. Before implementing in the clinical setting, the scope of the pathogen must be defined to decide the choice of sample type, the need for equipment and reagents, and the test methodology. An error-based approach should be used before assay validation to find the potential risks in the procedure. Clinical evaluation with real samples is important before implementing into the clinical setting.

**Figure 2 ijms-25-03333-f002:**
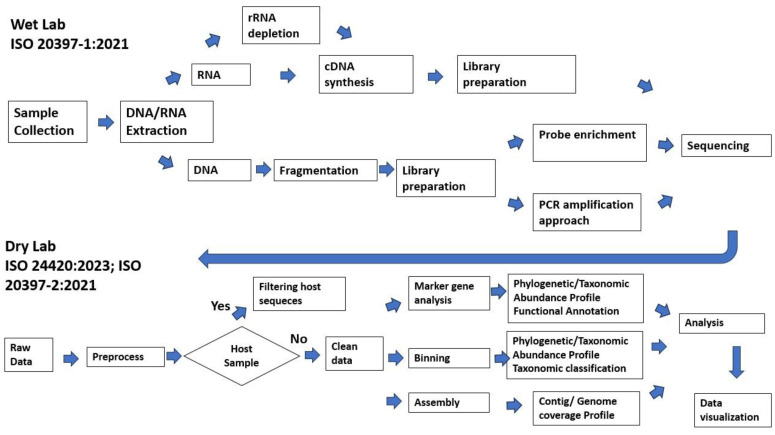
Overview of different sequencing approaches for infectious disease diagnosis. The wet lab workflow is based on ISO20397-1:2021, which included sample collection, DNA/RNA extraction, library preparation, target capture, sequencing, and bioinformatics analysis. The dry lab workflow is based on ISO24420:2023 and ISO20397-2. Three main metagenomic analysis approaches, including marker gene analysis, binning, and assembly, are described in this figure.

**Table 1 ijms-25-03333-t001:** Overview of the subject, action, and objective based on the 5M1E methodology using on NGS assay.

	Manpower	Machine	Material	Method	Environment	Measurement
**Subject**	Well-trained personnel, compliance with SOP	Automation extraction machine	Raw materials, reagent and samples	Procedures, method, and protocols	Environment conditions such as humidity, lighting, temperature, and cleanliness	Instrument, techniques, data, and tools
**Action**	Training, skill development, and monitoring of ongoing performance	Calibration, routine maintenance; adhering to equipment specification	Verification; sample acceptance criteria, reagent specification checking, storage and handling	Validation	Adequate controls, monitoring	Calibration, validation, and routine performance check
**Objective**	Minimize human-related variation, human error	Minimize variations in test results; ensure that the equipment is performing within acceptable limits and meets the required specifications; reduces the risk of errors and inaccuracies in test results	Ensures the integrity and traceability of materials, prevents contamination and degradation, and ensures high-quality materials are acquired	Ensure reliable and accurate result	Minimize variations caused by environmental factors	Ensure precise measurement

**Table 2 ijms-25-03333-t002:** An error-based approach using 5M1E methodology on NGS assay.

mNGS Procedure	Error/Risk	Consequence	5M1E	Means of Minimizing Risk
Sample collection	Improper storage [89] and handling of samples, such as higher freezing and thawing cycle	Degradation of nucleic acids Inaccurate representation of the microbial community present in the sample [89,90]	Method, Materials, Environment	Use sterile collection tools and containers Proper training of personnel involved in sample collection
Sample collection; nucleic acid extraction	Host DNA contamination/cross-contamination of samples	Influence the accuracy and reliability of sequencing results [91] Reduced sensitivity to detect microbial reads [92] Inaccurate classification of pathogenic communities [93] Misidentification of DNA samples and compromised experimental results [94,95]	Method, Machine, Materials, Environment	Reduce contamination by evaluating host depletion and extraction methods [83] Ensure sequence distribution and individual sequence errors are addressed with quality control measures [96] Calibration and decontamination of the machine
Nucleic acid extraction	Inadequate lysis of cells or tissues/non-verified reagent	Suboptimal DNA yield and compromised quality [97,98]	Materials	Implementing methods that selectively lyse host cells and/or separate specific organisms from clinical samples [99,100]
Nucleic acid extraction; library preparation	Improper sample preparation, such as incorrect sample loading or inadequate mixing	Inaccurate biological interpretations and impact on the reliability of the sequencing data [89,101] Affecting downstream application [102]	Methods	Proper training of personnel involved in the procedure
Nucleic acid extraction	Inhibitory Substances	Interfere with the downstream analysis and application [103]	Methods	Optimization of DNA extraction and purification procedures minimizes co-extraction of inhibitory substances with DNA [104] Evaluate the DNA preservation methods to avoid co-extraction of inhibitory substances [105]
Library preparation	High levels of DNA fragmentation	Reduced DNA yield and inaccurate quantitation, impacting the reliability of subsequent analyses [102,106]	Methods	Evaluate the incubation time for fragmentation
Library preparation	Index hopping	Erroneous conclusions and misinterpretations of the microbial community composition [107]	Methods	Incorporating unique dual-indexing strategies that assign unique index sequences to each sample [108]
Sequencing	Systematics error/sequencing error	Overestimation of gene and taxon abundance [109] Inaccurate gene prediction on short reads [110]	Machine, Methods	A simulation tool that incorporates error models based on explicit errors and coverage bias can help address sequencing errors [111] The development of a statistical method for functional profiling can allow sequencing errors to be estimated and possible functions to be detected in metagenomic data [112]
Bioinformatics analysis	Misinterpretation of results [113,114]; inadequate data preprocessing [115]; incorrect taxonomic or functional annotation [116]; failure to account for batch effects [117]; inadequate validation of findings [118]; lack of reproducibility [119]; overfitting or underfitting of models [120]; inadequate training data [121]	Impact the accuracy and reliability of the results	Manpower, Machine, Methods	Integrate comprehensive databases and update them regularly to incorporate newly discovered microbial species. Developing novel taxonomic classification algorithms and tools [122] Conduct of rigorous quality control measures as well as benchmarking and standardizing bioinformatics workflows Using validated statistical methods, coupled with careful consideration of data normalization and bias correction [123] Proper training of personnel

**Table 3 ijms-25-03333-t003:** Indicators of validation and QC metrics.

Measurement	Indicators of Validation	QC Metrics	Follow-Up Actions
Nucleic acid quality	DNA Purity	High DNA purity [125]: A260/A280: 1.88–1.94 Low DNA purity: A260/A280 < 1.6	Perform additional purification step, such as phenol-chloroform extraction or silica-column-based purification kits Optimize the extraction protocols to minimize contamination and improve the purity and quantity Perform additional purification step, such as DNase treatment or RNA cleanup kits Re-extraction
	RNA Purity	RNA quality acceptable range [126]: A260/A280: around 2 A260/A230: 2-2.2 RNA integrity number (RIN) analysis depending on different sample types Acceptable range 7–8	
Nucleic acid quantity	DNA/RNA concentration	Depend on the library preparation approaches: DNA: 20ug [127] 50–250 ng [128] RNA [129]: standard (1 ug); low (10 ng–100 ng); ultra-low (<1 ng)	
Library quality and quantity	Library size distribution and quality patterns Library concentration	For short-read sequencing Size distribution: 250–350 For long-read sequencing Size distribution: 10–25 kb (PacBio® HiFi sequencing) [130] 250 bp to 50 kbp (Nanopore sequencers (MinION, GridION, and PromethION) [131]	Ensure that fragment sizes within the narrow range of expected molecular weight match the measured fragment sizesPrevent adapter dimers, primer dimers (~150 bp), and molecular weight outside the expected range
Sequencing quality	Total run yield (Gb), sequencing cluster density (K/mm^2^), % reads passing filter, % bases ≥Q30, sequencing error rate, sequencing read length	For short-read sequencing: the sample base mass value shall be more than 20 if Q20 is greater than 90%; and the sample base mass value shall be more than 30 if Q30 is greater than 80% [42]	Resequencing should be conducted if sequencing QC cannot be achieved
Coverage depth	Sequence depth should be evaluated before sequence assembly, taking the sample complexity into consideration [42]	A sufficient level of sensitivity and specificity must be achieved in the regions of interest Short read: at least 30× coverage	Reanalysis of samples should be conducted if coverage thresholds exceed the validated range; an alternate method may be used if only local regions are affected
GC bias	GC content	No specific threshold for GC bias GC content <70% [132]	Optimization of library preparation [133] Implement filtering strategies [133] GC content correction [134] Evaluation of sequencing bias with different library preparation kit [135] Normalization techniques [136]
Base call accuracy	Phred quality score (Q), where Q = −10 log10(P) [137]	Good base call quality scores Phred score > 30 [138] Base calling: Short-read sequencing: >99.9% Long-read sequencing: ~90%	Each run should be monitored for quality scores and signal-to-noise ratios The results of low-quality scoring can lead to more false positive variant calls; therefore, repeat testing may be necessary
Duplication rate	Duplication rate	Maximum duplication rate should be defined for each assay.	Optimization of library preparation Adjust PCR conditions [42,43] Unique molecular identifiers (UMIs) [139] Filtering duplicate reads [140] Error correction algorithms [134]
Mapping quality	Mapping quality scores	Map quality parameters must be established during validation in order to filter out reads that map to nontargeted regions (insertion or indel) and uncertain bases (N characters) [42]	The mapping quality of each run must be monitored as non-specific amplification, off-target DNA capture, or contamination may result in poor results

**Table 4 ijms-25-03333-t004:** Software tools involved in bioinformatics analysis for metagenomics.

Bioinformatic Step	Software	Function	Reference
Preprocessing of Sequencing Reads	FastQC (v0.12.0)	Assesses the quality of raw sequencing data	
	Trimmomatic (v0.4)	Trimming and filtering reads to eliminate low-quality bases and adapter sequences	[177]
	Cutadapt (v3.4)	Efficiently removes adapter sequence	[178]
	DUST (v0.9)	De-replication of reads	
	QIIME (v2023.2)	Noise removal	
Fragment Recruitment to Reference Genomes	Bowtie2 (v2.5.3)	Mapping preprocessed reads to reference genomes, contamination removal	[179]
	BWA (v0.7.12)	Mapping low-divergent sequences against a large reference genome	
	BWA-SW (v0.7.12) and BWA-MEM (v0.7.12)	Mapping longer sequences (70 bp to 1 Mbp), share similar features such as long-read support and split alignment	[180]
	SAMtools (v1.19.2)	Manipulating and analyzing sequence alignment data, crucial for post-processing and downstream analysis of metagenomic data	[181,182]
De Novo Metagenome Assembly	MEGAHIT (v1.2.9)	Assembly of large and intricate metagenomic datasets using succinct de Bruijn graphs, providing a single-node solution for complex assemblies	[183,184]
	IDBA-UD (v1.1.3)	A specialized de Bruijn graph-based assembler designed for metagenomic sequencing data, aiding in the reconstruction of microbial genomes	[185]
	MetaSPAdes (v.3.13.0)	Advanced metagenomic assemblers that integrate information from multiple samples to improve accuracy and congruency	[186]
	QUAST (v5.0.2)	Evaluates genome/metagenome assemblies	[187]
Genome Binning	MaxBin (v2.0)	Binning tool that clusters contigs based on expectation-maximization algorithms, facilitating metagenomic data organization	[188]
	CONCOCT (v1.1.0)	The recovery of metagenome-assembled genomes in situations where the reference genome for a species of interest within a metagenome is unknown	[189]
	MetaBAT (v2.15)	Binning metagenomic contigs into genome bins based on sequence composition and abundance	[189]
Taxonomic and Functional Analysis of Genomes	Kraken 2 (v2.0.8 beta)	A highly accurate classifier for taxonomic sequences that rapidly assigns taxonomic labels to metagenomic sequences, allowing precise taxonomic analysis	[190]
	HUMAnN 3 (v3)	The tool provides insights into the functional potential of microbial populations based on metagenomic sequencing data	[191]
	MetaPhlAn 4 (v4)	This tool aids in taxonomic analysis of metagenomic shotgun sequencing data by profiling microbial communities	[192]
Metagenomic Assembly and Analysis	Anvi’o (v.2.1.0)	Analyze and visualize complex metagenomic datasets using an interactive platform for metagenomic analysis and visualization	[193]
	MEGAN-LR (v6.19.1)	A long-read version of MEGAN for taxonomic analysis and functional annotation of metagenomic data generated from long-read sequencing technologies	[194]

**Table 5 ijms-25-03333-t005:** Challenges and strategies for implementing NGS in infectious disease diagnosis.

Challenges	Strategies	Future Perspectives	References
Turnaround Time and Costs	Cost and turnaround time reduction	Development of more cost-effective NGS platforms Optimization of bioinformatics pipelines	[30,31]
Standardization and Quality Control	Develop universally accepted protocol or standard	Development of consensus standards through cooperation among researchers, industry stakeholders, and regulatory bodies	[34,35]
	Integration of automation	Integration of automation into the whole mNGS workflow	[231,232]
	Proficiency testing and reference materials	Expansion of proficiency testing programs for mNGS	[233]
	Standards and developments in bioinformatics	Availability and accessibility of rapidly evolving software to users Usage of metabarcoding and metagenomic bioinformatics	[140,234,235,236]
	Collaboration and open sharing	Supporting open science initiatives through funding mechanisms and academic recognition	[237]
	Technological advancements	The standardization and quality control of mNGS	[238]
Bioinformatics and Data Analysis	User-friendly tools	Development of user-friendly software and databases	[239,240]
	Data storage and privacy	Implementation of robust data privacy and security measures	[239,240]

## Data Availability

No new data were created or analyzed in this study. Data sharing is not applicable to this article.

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
