# Peer review of "Enhancing Clinical Utility: Utilization of International Standards and Guidelines for Metagenomic Sequencing in Infectious Disease Diagnosis"

_ijms, 2024, doi:10.3390/ijms25063333_

Round 1
Reviewer 1 Report
Comments and Suggestions for Authors
This article provides a comprehensive review of the strategies and challenges associated with implementing metagenomic next-generation sequencing (mNGS) in the diagnosis of infectious diseases. Overall, the manuscript is well-written. I have one minor suggestion:
Please consider adding more discussion about the clinical significance of mNGS in diagnosing infectious diseases and incorporating additional clinical studies to showcase its current practical utility in real-world scenarios.
Reviewer 2 Report
Comments and Suggestions for Authors
The authors reviewed application of NGS technology in clinical infectious disease diagnosis. NGS could provide sequence-based evidence for precision diagnosis. However, the manuscript lacks precision, e.g.,1) Lines 467-475, Sanger sequencing is the 1st generation sequencing, not 1st generation NGS; NGS is the 2nd generation sequencing, not 2nd generation NGS; and single molecule sequencing is the 3rd generation sequencing , not 3rd generation NGS. 2) Lines 395-401, controversy on automation of column-based methods. 3) Line 423, no carrier RNA should be used, not "a high concentration of carrier RNA should not be used". 4) Line 494, to date MiSeqDx is not the only one with FDA approval. 5) Lines 482-483, only nanopore long reads sequencing has real-time processing with advantage of speed; PacBio long reads sequencing has no advantage of speed. The authors need careful attention to the sentences throughout the manuscript.
The authors wrote "clinical interpretation of results and their implications for patient care should be provide concisely and clearly (Lines 590-591)". However, the manuscript is written long and abundant, not only in table but also in text. The table needs more concise, the whole manuscript could be cut in half by focusing on key points.
The classification of NGS application into WGS, mNGS and tMGS is problematic. It will be better to classify it with and without enrichment. The enrichment could be in levels of culture, PCR amplification, and hybridization; while no enrichment could be DNA or DNA/RNA based metagenomics and RNA based metatranscriptomics. Or, it can be classified to short-reads (Illumina & Ion Torent) and long-reads (nanopore & pacbio) using NGS platform.
Bioinformatics software and databases should be emphasized, with versions and update info recording, as well as the improvement of the pipeline on specificity, sensitivity, and LOD etc. While the authors mentioned machine learning in the text, though not practicable in clinical lab, they should have mentioned AI along with NGS application in future perspectives.
Regarding quality control and reference standards, have the authors considered about internal and external controls, which should be useful for LOD determination.
Comments on the Quality of English Languagelong and abundant in text and tables
Reviewer 3 Report
Comments and Suggestions for Authors
1. The abstract section needs to be refined to clearly indicate the significance of the study for next-generation sequencing.
2. In the abstract section, pathogen levels need to be clarified in the text.
3. How is NGS significant in identifying each microbe from mixed culture?
4. As the stati. analysis is significant in determining the sensitivity, specificity, positive or negative predictive value, it will be more significant if the author adds a paragraph for static analysis with the name of the used software.
5. In addition, optimizing processing conditions in sample preparation needs to be mentioned.
6. The bead-based method or the liquid extraction method, which method is most suited to NGS, and factors affecting the process, are discussed in this section.
7. DNA RNA extraction section: As plants, animals, bacteria, and fungi have different cell wall compositions, for NGS, each significant, specific method needs to be added.
Reviewer 4 Report
Comments and Suggestions for Authors
The review article by Kan C. M. et al. have systematically discussed the challenges and strategies during the implementation and validation of mNGS based on related guidelines and ISO standards.
The overall structure of the review is well-organized, providing a clear roadmap for readers and well written with clear description. However, some concerns should be addressed.
1. The title is so big and not aiming the content of review article. Please make this title more focus for the researchers.
2. There are several reviews articles those have already been talked about the Next-Generation sequencing and its application so how authors make this review article different then others.
3. The interval time of references in the introduction is too long and contains a maximum of older references, so it is suggested to quote the literatures in the last three to five years.
4. Authors need to write the discussion part in a descriptive way with explanation.
5. The language expression of the manuscript needs to be further simplified and polished.
Comments on the Quality of English LanguageThe language expression of the manuscript needs to be further simplified and polished.
Round 2
Reviewer 2 Report
Comments and Suggestions for Authors
Tables could be simplified more.
